

# Terracing Increases Organic Carbon Content in the Loess Plateau

Qinqin Wang[a,b,1], Yuanxiao Xu[c,1], Guofeng Zhu[a,b,*], Siyu Lu[a,b], Dongdong Qiu[a,b],

Yinying Jiao[a,b], Longhu Chen[a,b], Gaojia Meng[a,b], Rui Li[a,b], Xiaoyu Qi[a,b], Wenmin Li[a,b],

Ling Zhao[a,b], Yuhao Wang[a,b], Eenwei Huang[a,b], Wentong Li[a,b]

*[a] College of Geography and Environment Science, Northwest Normal University, Lanzhou 730070, Gansu, China*

*[b] Shiyang River Ecological Environment Observation Station, Northwest Normal University, Lanzhou 730070, Gansu, China*

*[c] State Key Laboratory of Herbage Improvement and Grassland Agro-ecosystems, College of Pastoral Agriculture Science and Technology, Lanzhou University, Lanzhou 730000, China*

*[1] These authors contributed equally to this work and should be considered co-first authors*

*[*] Corresponding author. Email: zhugf@nwnu.edu.cn.*

**Abstract**

**Aim:** Terracing is widely distributed in mountainous and hilly areas around the world and can be effective in inhibiting soil erosion, increasing soil moisture, improving soil quality and potentially having a positive impact on soil carbon pools.

**Methods:** To understand the impact of agricultural activities and ecological restoration measures on changes in soil carbon pools in terraced areas, we set up an observation system in typical terraces on the Loess Plateau and soil samples were



gathered from 0-100 cm depth in terraces (containing different crops and different
ecological restoration vegetation) and slopes.
**Results:** The results show that terracing can effectively increase soil organic carbon
(SOC) content (7.7 g·kg-1 in terraced cropland > 4.9 g·kg-1 in sloping cropland).
Changes in the organic carbon content of the terracing is mainly due to improvements
in soil and water conservation capacity and agricultural activities, loss of soil organic
carbon due to short-term abandonment and an increase in soil organic carbon due to
replanting of fruit trees and crops. The choice of tree species in afforestation policies
has also led to differences in soil organic carbon. Pinus tabuliformis Carr. has the
highest SOC content (9.8 g·kg-1).
**Conclusions:** The SOC content in 0-100 cm of terraced fields planted with wheat was
1.5 times higher than that of sloping fields planted with wheat. Compared with
sloping land, terrace construction significantly increased the SOC content of
cultivated land, especially in the top soil layer (0-30 cm), and converting some sloping
land into terraces would enhance the carbon sequestration capacity. This study has
significant implications for agricultural management and ecological restoration in the
terraced areas of the Loess Plateau and contributes to the development of rational
policies for carbon sequestration on arable land in terraced areas.
**Keywords:** Terracing; Soil Organic Carbon; Agricultural activities; Vegetation cover
**1 Introduction**

Soil is considered to be the second largest carbon reservoir next to the ocean



(Stockmann et al., 2013), and the soil carbon reservoir is crucial for the global carbon
cycle (Lal, 2004; Houghton, 2007). SOC is is a key element of the global carbon cycle
( Rossel et al., 2019) and serves as a significant indicator for assessing soil quality and
land productivity (Guillaume et al., 2021; Wang et al., 2012). To address climate
change and lower CO2 emissions for sustainable development, we should sequester
more carbon in the soil rather than releasing it into the atmosphere. This is already an
urgent and challenging issue for humanity (Bednar et al., 2021). Therefore,
understanding the variation of SOC content in different regions, especially in
anthropogenic landscapes, is of great importance to assess the carbon sequestration
potential of soils and mitigate climate change. The sequestration of large amounts of
$CO_2$ into soils can be achieved by the ways of land use and management practices
policies (Smith, 2012). However, SOC reserves have been declining worldwide (Jones
et al., 2005). Forest destruction caused by reclaiming farmland is the main reason for
global SOC consumption to date, resulting in increasingly serious ecological damage
(Lal, 2016). Hence, how to increase soil organic carbon to maintain soil quality,
restore ecology, ensure food security and reduce CO2 emissions is a matter of great
concern in the context of global warming.

Agricultural terracing is a crucial landscape engineering measure to reduce soil

erosion and maintain soil fertility and increase agricultural productivity (Doetterl et al.,
2012; Zhu et al, 2021), which is one of the ways to achieve sustainable agricultural
development. On the one hand, the conversion of terraces into slopes increases the
cultivated area significantly. On the other hand, it helps prevent erosion problems



(Arnáez et al., 2015) and effectively increases food production (Tarolli et al., 2014).
Terraces are widely distributed and have created environmental benefits in countries
in East Asia, the Mediterranean, and Southeast Asia (Wei et al., 2016). Many studies
have shown that terracing can intercept more than 80% of rainfall runoff and sediment,
and horizontal terracing can retain all rainfall to replenish soil moisture. The positive
benefits of carbon capture generated by terraces come from the collection of eroded
material for sloping soils. However, the conversion of natural vegetation to cropland
inevitably results in a reduction of biomass and therefore a significant loss of SOC
(Aguilera et al., 2018 and 2013). During the construction of terraces period, it is
inevitable stripping of topsoil and exposure of deep soil, and a large amount of new
subsoil covers the surface of the terraces. This severe soil disturbance may alter soil
organic carbon dynamics (Sidle et al., 2006), but the potential long-term benefits of
terrace construction are considerable (Chen et al., 2017). However, many terraces are
experiencing ridge damage and terrace collapse due to a lack of terrace maintenance
or land abandonment, which not only leads to reduce soil and water conservation
benefits but potentially increases erosion and carbon emissions (Arnáez et al., 2015;
Wen et al., 2020).
Agricultural land accounts for more than 30% of the global area and has great
potential for carbon sequestration and mitigation of global climate change (Sun et al.,
2010). Agricultural soils can be improved by implementing some regulatory
management practices to improve soil properties and further increase organic carbon
content (Lal et al., 2011). However, studies from different regions have shown a



global trend of decreasing organic carbon content in agricultural soils nowadays
(Bellamy et al. 2008; Heikkinen et al. 2013; Yli-Halla et al., 2018). How to increase
the SOC content of farmland is the key to promoting sustainable agriculture and
improving the carbon sink. In the Loess Plateau region, a large amount of arable land
has been converted to woodland and grassland through the implementation of
afforestation and reforestation policies, and these measures have increased the organic
carbon content of the soil (Rong et al., 2021). In the implementation of ecological
restoration, different vegetation types have different benefits in increasing soil carbon
pools (Hong et al., 2020). The agroforestry cropping pattern with other crops stores
more carbon than other traditional agricultural cropping patterns (Smith et al., 2022).
Nair et al. (2009) cited seven studies on soil carbon in tropical agroforestry systems
that show that this type of cropping certainly stores more carbon.
In the Loess Plateau, the erosion problem of sloping land has a great impact on
agricultural production (Ran et al., 2020). Terracing has become the most important
ways to solve the erosion problem on sloping lands. By the end of 2012, there were
37,100 km$^2$ of terraced fields on the Loess Plateau. With the further development of
terrace construction, changes in surface morphology and soil properties also lead to
dynamic changes in soil carbon pools, and such changes will lead to changes in soil
carbon storage capacity. Meanwhile, how the land use pattern of terraces and human
activity factors such as abandonment and crop type affect the soil carbon pool of
terraces still needs to be investigated. The implementation of afforestation policies in
terraced areas will also lead to changes in soil carbon pools. Consequently, we



gathered soil samples from terraces and slopes, including terraces with varying land
use and crop types, to investigate (1) the impacts of terrace construction on SOC in
the Loess Plateau region and (2) the effects of different vegetation cover and tillage
activities on the carbon sink capacity of terraces. This study will provide a reference
for terrace management and is also a guide for soil carbon sequestration.
**2 Data and Methods**
**2.1 Study Area**
The study area is the typical terrace construction region of the Loess Plateau
(Zhuanglang terracing) and the construction of this area began in the 1960s. By 2005,
14790 km² of terraces have been built, accounting for 95.3% of the total arable land in
the region. The structure of the terraces is mainly horizontal terraces. Zhuanglang
terraces belong to the loess hilly terrain area with gullies and complex topography,
and the elevation is between 1521m-1784m. The climate type is temperate continental,
with warm, humid summers and cold, dry winters. More than 60% of precipitation
occurs in summer and autumn (July - October), with an annual rainfall of 542mm and
an average annual temperature of 7.5℃. The dominant soil type in this area is fine
loessial soil, the natural vegetation is mainly herbaceous, shrubs, coniferous forests,
and locust trees, and the crops are wheat, maize, potatoes, and apple trees. Due to the
constraints of soil properties and irrigation water sources in this region, the growth of
crops depends on natural rainfall.



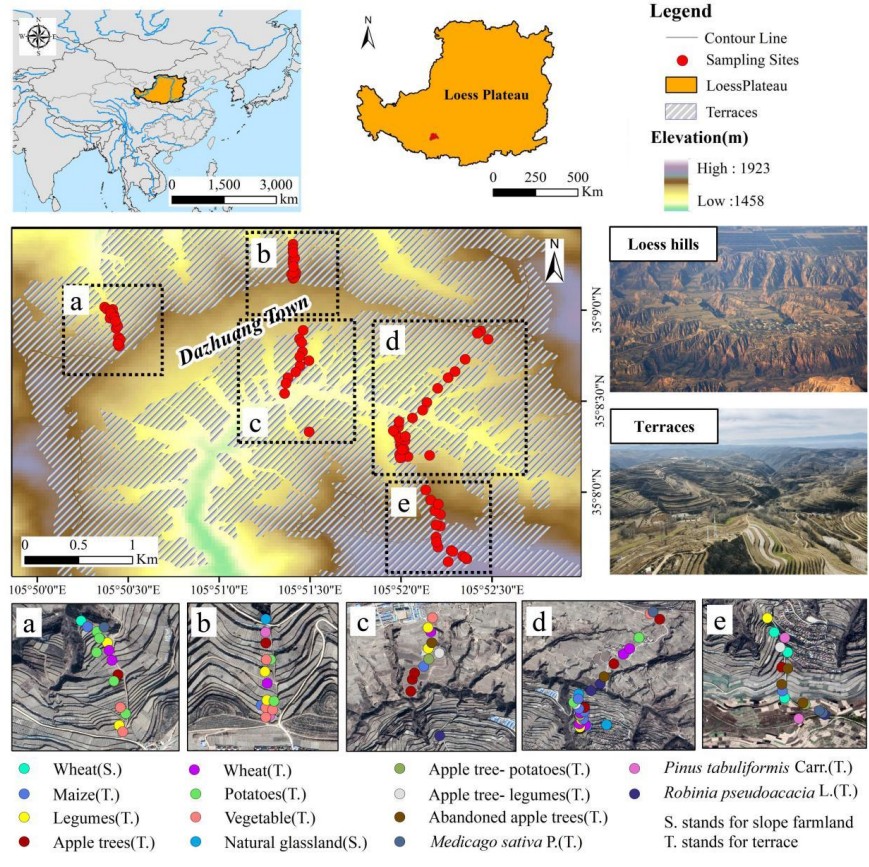


Fig.1 Study area

**2.2 Terraced soil sample collection**

We carried out soil sample collection at the terrace observation system in

Yangpota mountain, Dazhuang Town, Zhuanglang County, with the sampling date
being October 2020. The construction of terraces in this area started in 1964 and
based on interviews with local farmers, it was confirmed that all the agricultural fields
in the area had been constructed in 1991, and all the terraces were constructed 30
years ago. The main terracing structure built in the region is the horizontal terrace,





which is an agricultural field with stepped sections along contours on the slopes of
loess hills. The width of the terraces varies from 1.6 to 6 meters, and the height of the
terrace steps varies from 0.3 to 0.8 meters. The terraced slopes are slightly
counter-sloped to collect more precipitation and the slope ranges from 0% to 11%
(Chen et al., 2020).
We randomly set up 84 sampling sites in the study area, with 77 terraced
sampling sites and 7 slope sampling sites. Crop type, cropping pattern, and
agricultural abandonment all affect the soil carbon pool of the terraces, so the terraces
included sampling points for different cropping patterns of apple trees (9), number of
sampling points), vegetable (9), wheat (9), legume (9), potato (9), maize (9), apple
tree-legume (3) and apple- potato (3) (Appendix, Fig. A1). Five abandoned apple tree
terraces were included in the terraces and the apple trees were not removed from the
terraces and there was a large amount of weed growth. Three types of restored
vegetation were planted on the terraces: *Robinia pseudoacacia* L. (4), *Pinus*
*tabuliformis* Carr. (4) and *Medicago sativa* L. (4). The seven slope sites included four
wheat plantations and three natural grassland sites (Table 1).
Measuring SOC concentrations in the surface layer of the soil (10 or 20 cm)
alone does not imply soil changes due to tillage management, so we designed the
sampling depth as 1 m. Samples were collected at 0-10 cm, 10-20 cm, 20-30 cm,
30-40 cm, 40-50 cm, 50-60 cm, 60-70 cm, 70-80 cm, 80- 90cm, and 90-100cm
samples. A (2 × 2) m$^2$ sample square was randomly delineated in each identified
sample plot and sampled along the diagonal line. The soil samples collected three



times were mixed according to different soil layers after removing plant roots and
debris, and 840 mixed soil samples were finally obtained.




Table1 Number of different types of sampling plots, vegetation types, number of

samples, and soil property data

| Land types | Planting method | Vegetation types | Number of soil profiles | Soil depth(cm) | Soil texture fractions | | | Soil moisture(%) | | SOC(g•kg⁻¹) | |
|---|---|---|---|---|---|---|---|---|---|---|---|
| | | | | | Clay (%) | Silt (%) | Sand (%) | Mean | SD | Mean | SD |
| Terrace | Single vegetation | Wheat | 9 | 0-100 | 11.1 | 80.9 | 8.0 | 24.2 | 10.36 | 7.7 | 2.78 |
| | | Apple trees | 9 | 0-100 | — | — | — | 24.3 | 9.32 | 7.1 | 2.39 |
| | | Potatoes | 9 | 0-100 | — | — | — | 26.8 | 12.98 | 5.2 | 2.32 |
| | | Legumes | 9 | 0-100 | — | — | — | 23.2 | 15.21 | 4.5 | 2.10 |
| | | Maize | 9 | 0-100 | — | — | — | 21.4 | 10.35 | 7.7 | 2.48 |
| | | *Robinia pseudoacacia* L. | 4 | 0-100 | 10.8 | 79.4 | 9.8 | 18.2 | 5.92 | 8.0 | 2.77 |
| | | *Pinus tabuliformis* Carr. | 4 | 0-100 | 9.9 | 84.0 | 6.1 | 19.3 | 6.33 | 9.8 | 4.44 |
| | | *Medicago sativa* L. | 4 | 0-100 | 10.9 | 80.0 | 9.1 | 20.4 | 13.62 | 5.6 | 1.57 |
| | | Vegetable | 9 | 0-100 | — | — | — | 22.1 | 5.31 | 6.5 | 1.80 |
| | Multiple vegetation | Apple tree-legumes | 3 | 0-100 | — | — | — | 20.2 | 7.36 | 5.4 | 1.65 |
| | | Apple tree-potatoes | 3 | 0-100 | — | — | — | 22.7 | 10.68 | 6.7 | 1.77 |
| Sloping land | Single vegetation | Wheat | 4 | 0-100 | 10.0 | 78.0 | 12.0 | 21.3 | 13.44 | 4.9 | 1.07 |
| | | Grassland | 3 | 0-100 | 10.4 | 80.1 | 9.5 | 21.0 | 9.62 | 6.0 | 2.26 |
| Abandoned terraces | Multiple vegetation | Apple trees and weeds | 5 | 0-100 | 10.6 | 78.5 | 10.8 | 23.7 | 11.35 | 6.4 | 1.85 |

Note: "—" represents not measured. All soil attribute data values are average values, average soil
texture, soil moisture, and SOC of the 0-100 cm profiles, derived from 10 samples for each
profile (10-cm depth intervals)




**2.3 Experimental analysis and data statistics**
The collected soil samples were placed in sealed plastic bags and pre-weighed
aluminum boxes. The soil samples in the aluminum box were dried in the bake oven
at 105 °C for 24 hours to measure the soil moisture. After the samples were
completely shade-dried in the laboratory, gravels and plant roots were removed from
the samples using a sieve with a particle size of 2 mm. A 0.2 g soil sample was
weighed and the concentration of SOC was measured using a wet oxidation method
with dichromate (Nelson and Sommers, 1982 ). The soil texture, including sand, silt,
and clay content, was analyzed through a laser diffraction technique utilizing a
Mastersizer 2000 (Malvern Instruments, Malvern, England).
All data in this paper were analyzed by SPSS 21 statistical software. All
collected data underwent normality testing using the Kolmogorov-Smirnov test and
were assessed for homogeneity of variance with Levene's test, ensuring $P > 0.05$.
Comparative analysis of various sample point types was performed utilizing a
one-way ANOVA, with significance considered at $P \leq 0.05$. All data are expressed as
means ± standard deviation. Graphs were made using Origin 2021 software.
**3 Results**
**3.1 SOC characteristics of different land use types**
The SOC content of the abandoned apple tree terraces were lower than that of the
in-use apple tree terraces, but the difference was small at 0.7 g·kg$^{-1}$. The SOC content



of the wheat-grown sloping fields was significantly lower than that of the
wheat-grown terraces, with the SOC content of the terraces being 1.5 times higher
than that of the sloping fields, with a difference of 3.8 g·kg$^{-1}$ between the two (7.7
g·kg$^{-1}$ > 4.9 g·kg$^{-1}$). The SOC content of natural grassland was slightly higher (6.0
g·kg$^{-1}$ > 5.6 g·kg$^{-1}$) compared to planted grassland, although natural grassland with
weeds had not been terraced. Slopes with natural vegetation were higher in SOC (6.0
g·kg$^{-1}$) than those under cultivation (4.9 g·kg$^{-1}$) (Table 1).

The vertical variation of SOC at 0-100 cm depth varied significantly among land

use types (Fig. 2). Except for the abandoned land, all land use types showed an
irregular decreasing change pattern from the surface layer to the deep layer of the soil.
In the 0-10cm soil layer, the highest SOC content was found in terraces planted with
wheat and the lowest SOC content was found in terraces planted with *M. sativa*.
Terraces planted with fruit trees had significantly higher SOC content at 80-100cm
depth than terraces planted with crops, *M. sativa,* and natural grassland. The vertical
variation of SOC in sloping fields and terraces planted with wheat was consistent,
with their greatest SOC content occurring at 0-20 cm depth and their smallest SOC
content at 90-100 cm depth. In the abandoned terraces, SOC varied between 0-80 cm,
with the smallest SOC content occurring in the 80 cm soil layer.



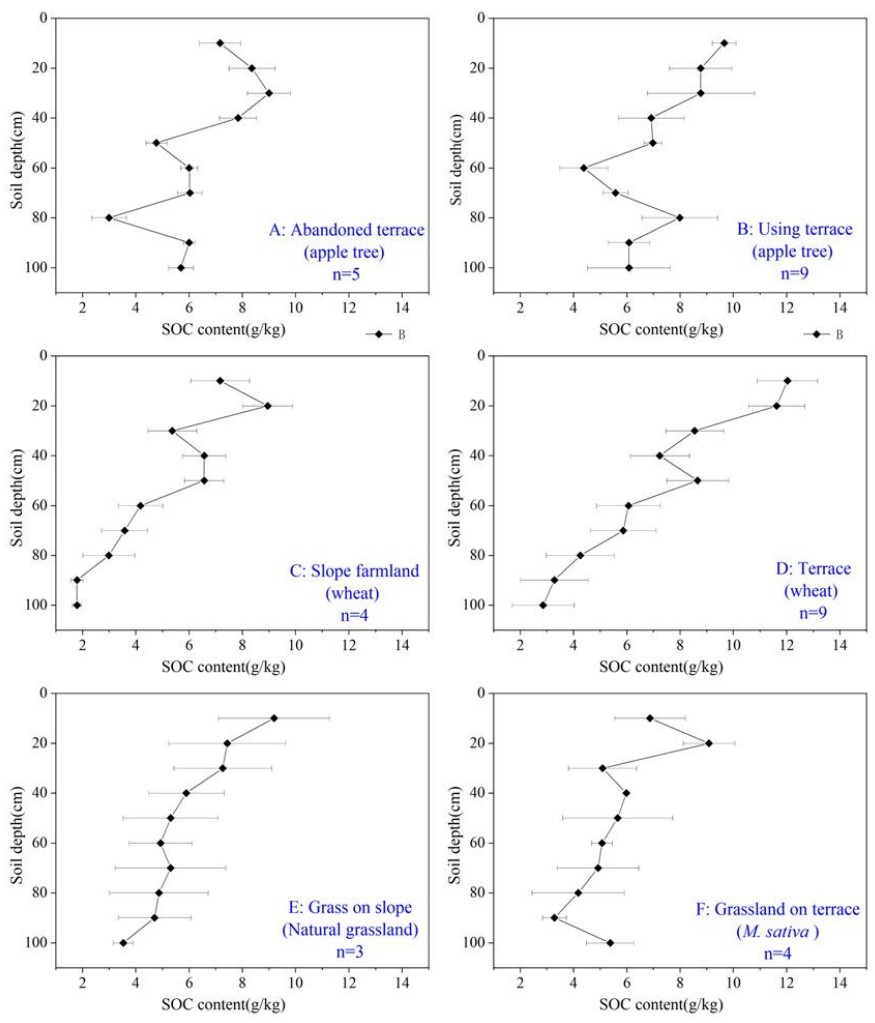

Fig.2 SOC Vertical variation of SOC content by land use type

Bars denote the standard deviation of the mean, n represents the number of soil profiles.

## 3.2 Characteristics of SOC in terraces of different planting patterns

Differences in SOC were smaller in terraces planted with apple trees and greater between terraces planted with a single crop. Among all crop types, legumes had the lowest SOC content (4.5 g·kg⁻¹) and maize had the highest SOC content (7.7 g·kg⁻¹).



The SOC content of concurrent apple tree - legumes and apple tree - potatoes was
higher than that of legumes and potatoes grown alone (5.4 g·kg$^{-1}$ > 4.5 g·kg$^{-1}$ and 6.7
g·kg$^{-1}$ > 5.2 g·kg$^{-1}$) (Table 1).

The vertical distribution of SOC content across a depth of 0-100 cm showed a

consistent decline from the surface soil to the deeper layers for all crops. The SOC
content (15.1 g·kg$^{-1}$) of wheat cultivated terraces was the highest among all crops at
the soil surface (0-10 cm). Terraces planted with apple trees, maize, or wheat had
higher SOC content in deeper soils (30-100 cm). Beans and potatoes had lower SOC
content at 50-100 cm depth than terraces planted with other crops. The difference in
SOC content between potato terraces planted alone and apple tree-potato terraces at
0-20 cm depth was not significant. However, below 20 cm depth, the SOC content of
the apple tree-potatoes combination was significantly higher than that of the terraces
planted with potatoes alone. This difference is also reflected in the legumes and apple
tree-legumes (Fig.3).






Fig.3 Vertical distribution of SOC content in different crop types





Bars denote the standard deviation of the mean, n represents the number of soil

profiles.

**3.3 Characteristics of SOC in terraces with different ecologically restored**


**vegetation**


Comparing the SOC characteristics of the three types of ecologically restored
vegetation after terracing, the average SOC content of terraces planted with trees
0-100 cm was higher than that of terraces planted with forage (Table 1). This
difference varied at different depths, with the SOC content of alfalfa being
significantly lower than that of the two trees in the 0-10 cm soil surface layer,
becoming smaller in the 10-20 cm depth, but increasing again in the 20-60 cm depth.
At 70-100 cm depth this difference became smaller and the SOC content between the
three vegetation species became close.
The difference in SOC between different silvicultural species was higher in *P.*
*tabuliformis* than in *R. pseudoacacia* at 0-100 cm depth, with a difference of 1.83
g·kg$^{-1}$. The significant difference in SOC between the two species was mainly at 0-70
cm depth, where *P. tabuliformis* had a higher SOC content than *R. pseudoacacia*. The
difference in SOC content between the two species became smaller at the depth of
70-100 cm, and the SOC content of *R. pseudoacacia* was slightly higher than that of *P.*
*tabuliformis* (Fig.4).



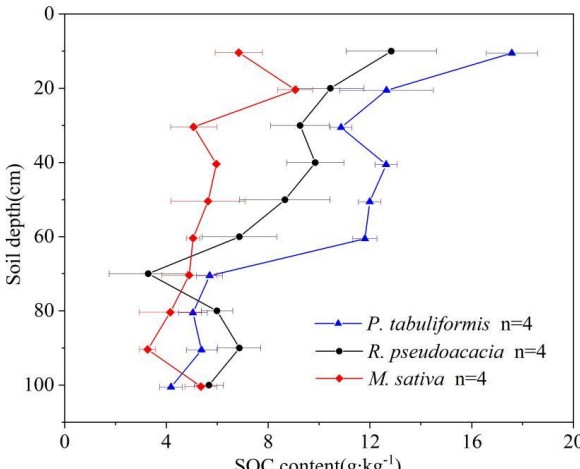


Fig.4 SOC vertical variation of different ecologically restored vegetation.

Bars denote the standard deviation of the mean, n represents the number of soil

profiles.

**4 Discussion**
**4.1 Effect of terrace construction on SOC**

In the Loess Plateau area, the average SOC content of terraces 0-100cm is 1.4

times higher than that of sloping farmland (Table 1). In Zhang et al. (2013), the SOC
stock at 0-100 cm depth was 4.97 kg·m⁻² in terraces and 3.09 kg·m⁻² in sloping fields,
which is 1.6 times higher than the soil organic carbon stock in sloping fields, which is
consistent with the results of this study. Terracing is considered an important practice
to prevent water erosion and minimize the loss of SOC (Nie et al., 2017). There is a
positive feedback relationship between soil moisture and soil carbon (Green et al.,
2019). Horizontal terraces are the most widespread terraces in the Loess Plateau,



which has changed the surface morphology, increased the time of rainfall storage at
the surface, and enhanced soil moisture in rain-dependent farming regions of the
Loess Plateau (Xu et al., 2021). Ecological stress caused by soil water deficit leads to
a decrease in biomass and net primary plant productivity. Conversely, an increase in
soil water has a positive effect on biological growth (McDowell et al., 2015). The
interception of precipitation on the terrace surface provides water for plant growth,
increases plant biomass, and increases the organic matter put into the soil, thus having
an impact on the SOC content. The interception of precipitation by the terraces means
that precipitation will no longer carry large amounts of fine soil particles from the soil,
which will increase the content of clay particles in the soil. Soil clay particles have a
larger specific area, which can adsorb more soil organic carbon and enhance the
accumulation of organic carbon (Post et al., 1982). Compared to sloping land, terraces
have a higher content of both clay and silt in the soil. The terraces therefore further
contribute to carbon accumulation in the terraces by protecting the fine particles in the
soil. In a study on the Loess Plateau, the SOC content of 0-100 cm in unterraced date
palm orchards was 2.6 g·kg$^{-1}$, which was lower than the soc content of terraced
orchards. This evidence further demonstrates the positive effect of terracing on soil
organic carbon sequestration (Gao et al., 2017).

The SOC varies significantly in terms of the amount of plant and animal residues

entering the soil and the depth of the soil under agricultural cultivation (Koga et al.,
2020). The impact of agricultural activities on the surface soil levels was stronger
compared to the deeper soil levels (Li et al., 2020). In this study, we observed a



significant increase in SOC in terraces than in sloping lands, particularly in the 0-30
cm soil layer (Fig.5). Post-terracing, SOC sequestration in deeper soils lagged behind
that in surface soils. Furthermore, the rate of SOC change was more pronounced in
the surface layer (0-20 cm) compared to the deeper layer (20-100 cm). Precipitation in
the region is limited and cannot replenish deep soil water, and the erosion of
precipitation on the slope surface also mainly takes away the top soil layer. Therefore,
the soil and water conservation effect brought by terrace construction is limited, so for
the soil depth increases, this effect will become smaller. The impact of terracing on
SOC sequestration diminishes as soil depth increases (Deng, Liu, and Shangguan.,
2014). As soil depth increases, the water stored in the terraces cannot penetrate deeper
soils and deeper soils will maintain their properties. Therefore, the management and
conservation of terrace topsoil are important to ensure local food production and
enhance the carbon sink function (Li et al., 2014).

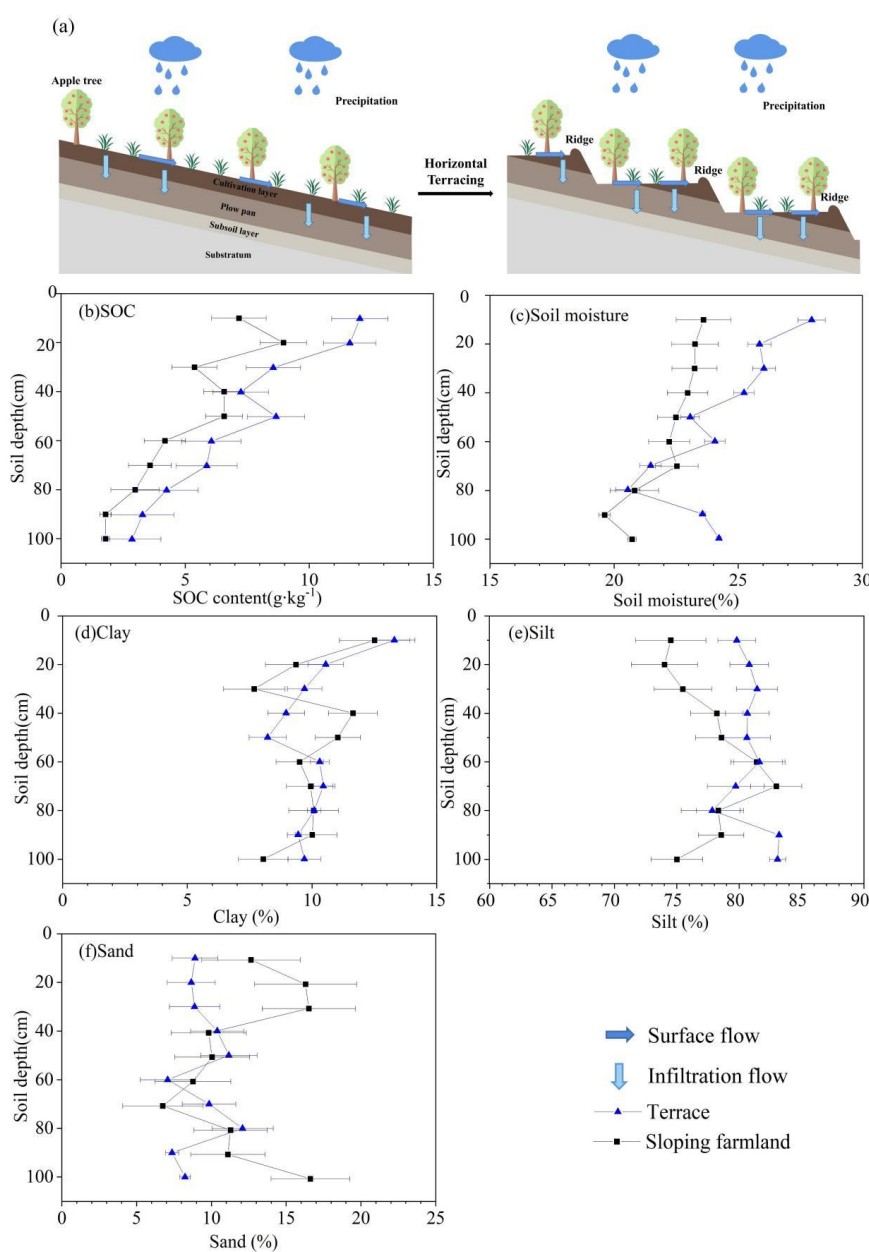


Fig.5 Effect of terrace construction on SOC, soil moisture, and soil grades.

(a): Variation in surface morphology by terrace construction; (b): variation in SOC
content; (d): variation in soil moisture; (d), (e), and(f): variation in soil grades. The





number of profiles is 9 for terraces and 4 for sloping fields. Bars denote the standard

deviation of the mean.

**4.2 Effect of terraces abandonment on SOC**

As in other parts of the world, industrialization and urbanization have led to a

large population flock from rural to urban areas as in China, resulting in the
abandonment of a large number of productive potential farmlands (Wiesmeier et al.,
2012; Cai et al., 2016). Moreover, a large amount of arable land in rain-fed
agricultural areas in the Loess Plateau region has been abandoned due to water
resource constraints or due to declines in soil fertility (Cao et al., 2020). Secondary
succession of vegetation after terraces abandonment leads to an increase in soil carbon
content as a potential pathway for climate change mitigation (Bell et al., 2021). When
the terraced fields were abandoned in this research, the SOC content of the abandoned
terraces was lower than that of the terraces in use. This is caused by the short
abandoned time. To produce significant environmental benefits, the land must remain
abandoned for an extended period to accumulate substantial amounts of both plant
biomass and the species that constitute intact ecological communities. This process
can take decades to reach levels of carbon sequestration or biodiversity comparable to
those of undisturbed ecosystems (Crawford et al., 2022; Poorter et al., 2016). Due to
the limited water resources available in semi-arid areas, a longer natural or assisted
recovery time is required. Therefore, the duration of land abandonment is a crucial
factor influencing the dynamic changes SOC (Djuma et al., 2020; Badalamenti et al.,



2019). In related studies in other regions, soil carbon stocks increased by 13% and
16% in cropland abandoned for 15 and 35 years, respectively (Novara et al., 2014).
With the abandonment of disposal time extended, vegetation types gradually
transition to grassland, scrub, and forest and the death of plants and animals return to
the soil as organic matter, increasing the number of soil aggregates and further
increasing the carbon content of the soil (Liu et al., 2020). Therefore, ecological
restoration of newly abandoned terraces should be carried out as soon as possible.
After short-term abandonment, the terraced fields showed a special change pattern at
different depths in this study. SOC content first decreased and then increased with
increasing soil depth. The decrease in surface SOC was controlled by the decrease in
agricultural fertilizer inputs, while the increase in deep SOC was caused by the
inability to utilize deep soil nutrients due to the death of crop roots.
**4.3 Effect of vegetation type and planting patterns on SOC in terraces**
Vegetation types can influence SOC by modifying the soil's physicochemical
structure and altering both the input and decomposition rates of SOC (Du et al., 2022;
Wiesmeier et al., 2012; Wan et al., 2019). Our study demonstrated that, compared to
terraced fields, the SOC content of afforested land at a 0-100 cm depth was higher and
that the forest litter biomass was more than that of farmland, which was the main
reason for this difference. Planted forest land reduces soil temperature, soil moisture
evaporation, and soil erosion while increasing the quantity and quality of organic
matter input to compensate for carbon decomposition from crop cultivation (Liu et al.,





2020). The afforested land is terraced forests, and the effect of preventing soil erosion
is more significant. Some study shows that the SOC in immature forests (10 years old)
is 17.91% higher than that in terraced cropland. The SOC concentration of a
30-year-old forest is significantly higher than that in other land covers (Xin et al.,
2016). These studies further proved the carbon sequestration effect of reforestation.
Due to the problem of ecological degradation and soil erosion, various ecological
measures have been taken in the Loess Plateau area, such as returning farmland to
forest and grass and planting trees (Hong et al., 2020). Considering the climate and
soil quality factors, the main species selected in the Loess Plateau region are
drought-tolerant types of trees, and the carbon accumulation effect of different species
selection also differs significantly (Li et al., 2018). *P. tabuliformis* has a higher SOC
content than *R. pseudoacacia*, especially in the 0-50 cm soil layer. The pine species
selected in this region is larch, with the arrival of winter a large number of pine
needles and fruits are into the soil, increasing the input of organic matter in the
surface layer so that the SOC content of pine forests is higher in the surface layer of
the soil (0-10 cm). In humid areas, some studies also show that the soil organic carbon
density of fir conifer forests is the largest among the different 11 middle forest
vegetation types (Chen et al., 2007). Other studies have shown that tree species such
as *P. koraiensis*, *L. gmelinii,* and *P. tabuliformis* increase soil organic carbon stocks
more as silvicultural species (Hong et al., 2020). The biomass of the herbaceous
plants themselves is much lower than that of trees, and the limited amount of organic
matter entering the soil, and the fact that *M. sativa* is mainly used as a source of



fodder for the animals raised by farmers in the region, leads to a lower SOC content in
terraces planted with *M. sativa* than in those undergoing afforestation.
The SOC content of grassland at a depth of 0-100cm is lower than that of
farmland. Although the grassland has organic matter after the withered herbs enter the
soil, the main planting type in the terrace area is apple trees, and a large amount of
fruit tree leaves will also enter the soil. Grassland is a sloping land that has not been
terraced, leading to slope erosion that removes a significant amount of organic matter
from the soil surface. As a result, the SOC content in grassland is lower than in
terraced fields (Fig.2). The ecological advantages of sequestering SOC and enhancing
soil fertility could be significant, largely thanks to the widespread implementation of
reforestation and various land use strategies in terraced fields across China and
numerous other mountainous areas globally (Hong et al., 2020).
Crops may differ in their ability to increase SOC content due to differences in
their photosynthetic capacity and root characteristics (Wegener et al., 2015). The
pattern of intercropping in this area is typical of Agroforestry systems (AFS), where
other crops are planted between the rows of apple trees. The SOC content of apple
trees in combination with other crops was higher than in monocultures, especially in
the lower and middle layers of the soil (30-100 cm). The amount of tree litter and root
decomposition are important reasons for this (Pardon et al., 2017). The fallen leaves
of fruit trees and some rotting apples are not removed, and these organic materials
decompose to replenish SOC after entering the soil. In addition, carbon input can be
achieved by decomposing (fine) tree roots and root secretions (Nair et al., 2009). For



soils below 30 cm depth, tree roots produce an important role in the accumulation of
soil organic carbon. When potato or legume crops are harvested, all the fruit and plant
roots are removed and these lands will be tilled to grow other crops, so the input of
organic matter is very limited. Agroforestry systems increase the distribution of roots
in the soil and increase the recalcitrant compounds which slow the rate of
mineralization through the input of organic matter (Recous et al., 2008).
**4.4 Study limitations**

The results of the study are based on field data collected over a relatively short

period of time. Due to the complexity of field conditions, the number of soil profiles
in some of the comparative studies in the sampling frame design was not entirely
consistent, and future studies will need to expand the study area to achieve balanced
sampling. This study examined differences in SOC at individual time points, and
follow-up assessments are needed to confirm long-term trends. Factors such as soil
bulk density, soil ph, root biomass, fertilizer management, and tillage practices also
affect soil organic carbon in terraced areas, and more indicator measurements and
studies are necessary. We need to do more work to understand the SOC characteristics
of terraced agricultural areas and how to better utilize the terraces for carbon storage
and realize the economic and ecological value of terraces.
**5 Conclusions**

The results showed that the SOC content in 0-100 cm of terraced fields planted





with wheat was 1.5 times higher than that of sloping fields planted with wheat.
Compared with sloping land, terrace construction significantly increased the SOC
content of cultivated land, especially in the top soil layer (0-30 cm), and converting
some sloping land into terraces would enhance the carbon sequestration capacity.
Abandonment, vegetation type and planting structure affect the SOC of terraces.
planting other crops between rows of apple trees can increase the SOC content. Since
vegetation restoration takes a long time, short-term abandonment will lead to a
decrease in terrace SOC, and some abandoned terraces can be planted with ecological
restoration vegetation. Among the ecologically restored plant species, the vegetation
with the highest SOC content is Pinus oleifera. The SOC content of terraces planted
with artificial forage is lower than that of natural grassland, so it is necessary to
protect the natural grassland left behind and choose tree species with better ecological
benefits when planting trees. In the face of China's huge food pressure and the goal of
increasing carbon sinks to mitigate global climate change, terraces have significance
and importance. Continuous strengthening of terraces management will give full play
to their carbon sequestration role.
**Data Availability Statement**

The data that support the findings of this study are available on request from the

corresponding author, soil organic carbon data are not publicly available due to
privacy or ethical restrictions.



**Acknowledgments**

This research was financially supported by the National Natural Science Foundation of China(42371040, 41971036), Key Natural Science Foundation of Gansu Province(23JRRA698), Key Research and Development Program of Gansu Province(22YF7NA122), Cultivation Program of Major key projects of Northwest Normal University(NWNU-LKZD-202302), Oasis Scientific Research achievements Breakthrough Action Plan Project of Northwest normal University(NWNU-LZKX-202303).

**Conflict of Interest Statement**

The authors declare no conflicts of interest.

**Author contributions**

Guofeng Zhu and Qinqin Wang conceived the idea of the study; Siyu Lu, Xiaoyu Qi and Ling Zhao analyzed the data; Dongdong Qiu, Longhu Chen and Rui Li were responsible for field sampling; Qinqin Wang and Yuanxiao Xu participated in the experiment; Yinying Jiao, Gaojia Meng and Wenmin Li participated in the drawing; Qinqin Wang wrote the paper; Yuhao Wang, Wentong Li and Eenwei Huang checked and edited language. All authors discussed the results and revised the manuscript.



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





## Appendix

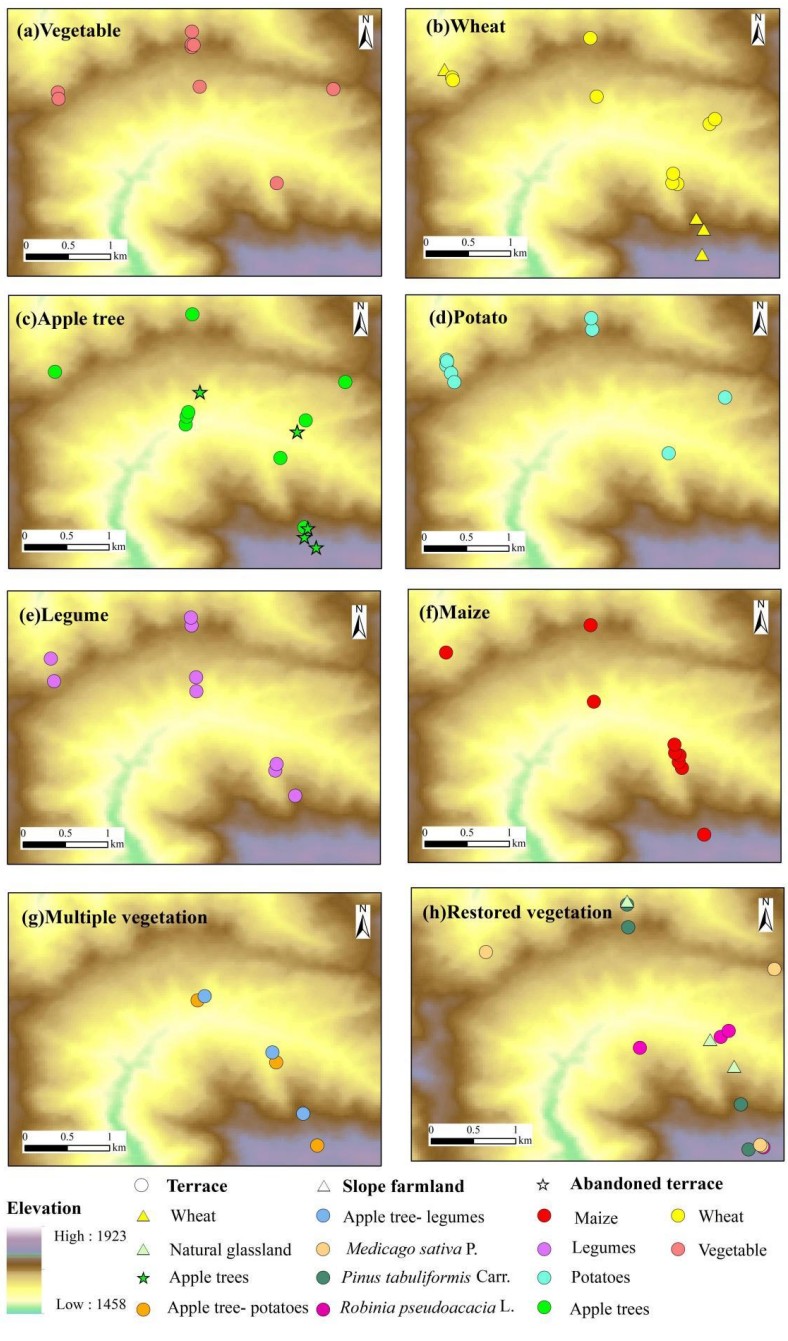

Fig.A1 Detailed distribution of randomly placed sampling points in the study area. These sampling



points cover various crop types, including apple trees, vegetable, wheat, legume, potato,
maize, apple tree-legume, and apple-potato. Additionally, the figure also displays the
distribution of sampling points under different planting patterns, specifically terraced
sampling points, sloping land sampling points, and abandoned terraced sampling points.