# Peer review of "Terracing Increases Organic Carbon Content in the Loess Plateau"

_EGUsphere, 2024_

## Author Comment (AC1)

RC1: 'Comment on egusphere-2024-2047', Anonymous Referee #1, 30 Aug 2024 rep

Land-terracing activities have been widely developed in mountainous and hilly areas mainly in China and the world other places in order to inhibit soil erosion, increase soil moisture, and improve the soil qualities. This study seems to be a systematic work to well understand the impact of agricultural activities and ecological restoration measures on changes in soil carbon pools in terraced areas. as my assessment, this study has significant implications for agricultural management and ecological restoration in the terraced areas of the Loess Plateau in China and even instruct for the world other countries. Thus, I recommend acceptance for publication and also encourage authors well address and improve the shortcomings for the whole text in consideration on my suggestions below:

Response:Thank you very much for your high evaluation of our research work. We are pleased to receive your support and suggestions. We will seriously respond to and incorporate your recommendations to further enhance the quality of this research paper. We have limited the "Loess Plateau" section to the context of China in the full text to avoid any ambiguity. Additionally, we have included a brief introduction to the history of terracing in the "Loess Plateau" region in the introduction. We also explored the correlation between changes in soil carbon storage and the timing of terracing, aiming to reveal the impact of long-term terracing activities on the soil carbon pool in this area. In the Data and Methods section, we emphasized the representativeness of the study area. We added a discussion on the potential impact of climate change on changes in soil organic carbon. Regarding the issue of lower SOC content in abandoned orchard terraces compared to actively used terraces, we provided a detailed explanation of the reasons for this outcome. Finally, we highlighted the innovation and main contributions of this study at the end of the introduction to emphasize its novelty compared to existing research.

1) In the whole text, many places about Loess Plateau should be limited in China.

Response:Thank you very much for your suggestion. You reminded us that the scope of the "Loess Plateau" should be confined to China, which is very important for clarifying the research context and improving the accuracy of the article. We carefully reviewed the entire text and made revisions to the relevant content. The revised content is as follows:

**Abstract**

Terracing is widely used in mountainous and hilly areas worldwide to control soil erosion, increase soil moisture, and improve soil quality, potentially impacting soil carbon pools. This study investigates how agricultural activities and ecological restoration measures affect soil carbon pools in terraced areas of the Chinese Loess Plateau.

**Introduction**

Terraces are widely distributed and have created environmental benefits in countries in East Asia, the Mediterranean, and Southeast Asia (Wei et al., 2016). The terraced field construction in the Loess Plateau region of China has a long history. China has historically emphasized soil and water conservation as well as agricultural production, gradually developing and refining terracing techniques in this process. By 2005, the area had established terraced fields covering 14,790 square kilometers, accounting for 95.3% of the total arable land (Ma et al., 2015). This long history of terrace construction may influence the current processes and storage of soil organic carbon in the region. Terraced fields that were established earlier typically have higher soil organic carbon content. This is because these older terraces have undergone a longer process of soil organic matter accumulation, resulting in the accumulation of more fresh organic matter, which enhances soil organic carbon storage (Deng et al., 2014; Rong et al., 2021). In contrast, newly constructed terraces often have lower organic carbon content due to the removal of topsoil, which requires a longer time for soil organic matter reconstruction (Sidle et al., 2006). In the Chinese Loess Plateau region, a large amount of arable land has been converted to woodland and grassland through the implementation of afforestation and reforestation policies, and these measures have increased the organic carbon content of the soil (Rong et al., 2021).

**Study Area**

The study area is the typical terrace construction region of the Chinese Loess Plateau (Zhuanglang terracing) and the construction of this area began in the 1960s. By 2005, 14790 km2 of terraces have been built, accounting for 95.3% of the total arable land in the region.

2) Please briefly introduce the history of land terracing activities in the Loess Plateau. Different from other countries, Chinese terracing policy has a long history. I am wondering whether there has a correlation between the soil carbon variation and terracing time or not.

Response:Thank you very much for raising this constructive question. You pointed out that the history of terracing in China's Loess Plateau region differs from that of other countries, which is an important context that merits further exploration. We have added the following content in response to this point:

The terraced field construction in the Loess Plateau region of China has a long history. China has historically emphasized soil and water conservation as well as agricultural production, gradually developing and refining terracing techniques in this process. 85% of the terraced fields in the Loess Plateau region were formed between 1950 and 2000, by 2005, the area had established terraced fields covering 14,790 square kilometers, accounting for 95.3% of the total arable land (Ma et al., 2015). This long history of terrace construction may influence the current processes and storage of soil organic carbon in the region. Terraced fields that were established earlier typically have higher soil organic carbon content. This is because these older terraces have undergone a longer process of soil organic matter accumulation, resulting in the accumulation of more fresh organic matter, which enhances soil organic carbon storage (Deng et al., 2014; Rong et al., 2021). In contrast, newly constructed terraces often have lower organic carbon content due to the removal of topsoil, which requires a longer time for soil organic matter reconstruction (Sidle et al., 2006). Since 2000, in the Chinese Loess Plateau region, a large amount of arable land has been converted to woodland and grassland through the implementation of afforestation and reforestation

policies, and these measures have increased the organic carbon content of the soil (Rong et al., 2021).

3) Widely terracing lands have been distributed almost all over the Loess Plateau and covering many kinds of climate types. In this text, a small county of Zhuanglang was selected as research area. Please give an evaluation on its representativeness of terracing lands. I think the climate changes should also play important roles for SOC variation even the similar terracing condition.

Response:We have carefully considered and revised the suggestions you provided. The rationale for selecting the Zhuanglang terraced fields as a representative example of terraced agriculture is explained in detail below:

Zhuanglang County is located in the central region of the Loess Plateau, characterized by typical geographical and climatic conditions, making it a representative area for studying terraced farming systems on the plateau. Construction in this region began in the 1960s. By 2005, a total of 14,790 square kilometers of terraced fields had been cultivated, accounting for 95.3% of the region's arable land. The terraced fields are primarily horizontal terraces, with a relatively complete system that effectively reflects the characteristics of traditional terraced agriculture in the area. Zhuanglang terraces belong to the loess hilly terrain area with gullies and complex topography, and the elevation is between 1521m-1784m. The climate type is temperate continental, with warm, humid summers and cold, dry winters. More than 60% of precipitation occurs in summer and autumn (July - October), with an annual rainfall of 542mm and an average annual temperature of 7.5℃. The dominant soil type in this area is fine loessial soil, the natural vegetation is mainly herbaceous, shrubs, coniferous forests, and locust trees, and the crops are wheat, maize, potatoes, and apple trees. Due to the constraints of soil properties and irrigation water sources in this region, the growth of crops depends on natural rainfall. In recent years, however, there has been a significant increase in abandoned terraces, highlighting the widespread issue of terrace abandonment in the Loess Plateau area.

We believe that your viewpoint regarding the significant role of climate change in the

variations of soil organic carbon is very accurate. However, the time frame of our observations is too short to effectively analyze the relationship between changes in soil organic carbon and climate change. We will analyze this aspect in the discussion section.

**4.2 Effect of terraces abandonment on SOC**

As in other parts of the world, industrialization and urbanization have led to a large population flock from rural to urban areas as in China, resulting in the abandonment of a large number of productive potential farmlands (Wiesmeier et al., 2012; Cai et al., 2016). Furthermore, climate change induced extreme weather events such as drought and heavy rainfall can also accelerate soil erosion and loss of soil organic carbon in the abandoned terraces (Lal, 2004). We measured the physicochemical properties of the soil in terraced fields with different usage statuses (Table 2). The results show that the soil bulk density in abandoned terraces is significantly higher compared to the actively used ones. This increased bulk density may lead to reduced soil aeration, thereby inhibiting the decomposition of organic matter. Furthermore, the soil pH in abandoned terraces has also decreased, which may affect the stability of organic matter. However, climate change can also impact the vegetation succession on abandoned terraces, which in turn affects the soil organic carbon dynamics (Davidson & Janssens, 2006). When the terraced fields were abandoned in this research, the SOC content of the abandoned terraces was lower than that of the terraces in use. This is caused by the limited abandoned time. Abandoned terraces may have accumulated a significant amount of organic matter during their previous use. However, due to a lack of fertilization now, this organic matter is gradually being mineralized and decomposed, which reduces the soil organic carbon (SOC) content (Lal, 2004; Wiesmeier et al., 2019). In contrast, terraces that are still in use maintain higher SOC levels thanks to continual fertilization (Nardi et al., 2004). Additionally, the abandoned terraces are more susceptible to climate change induced soil disturbance and erosion, leading to the loss of nutrient-rich topsoil, which further decreases SOC levels (Zhao et al., 2013). Our data also shows that the surface soil organic carbon (SOC) content in abandoned terraced fields (0-15 cm) is significantly

lower than that in actively used terraced fields, which may be related to higher soil bulk density, lower pH, and surface soil erosion (Table 2). To produce significant environmental benefits, the land must remain abandoned for an extended period to accumulate substantial amounts of both plant biomass and the species that constitute intact ecological communities. This process can take decades to reach levels of carbon sequestration or biodiversity comparable to those of undisturbed ecosystems (Crawford et al., 2022; Poorter et al., 2016). Due to the limited water resources available in semi-arid areas, a longer natural or assisted recovery time is required. Therefore, the duration of land abandonment is a crucial factor influencing the dynamic changes SOC (Djuma et al., 2020; Badalamenti et al., 2019). In related studies in other regions, soil carbon stocks increased by 13% and 16% in cropland abandoned for 15 and 35 years, respectively (Novara et al., 2014). With the abandonment of disposal time extended, vegetation types gradually transition to grassland, scrub, and forest and the death of plants and animals return to the soil as organic matter, increasing the number of soil aggregates and further increasing the carbon content of the soil (Liu et al., 2020). Therefore, ecological restoration of newly abandoned terraces should be carried out as soon as possible. After short-term abandonment, the terraced fields showed a special change pattern at different depths in this study. SOC content first decreased and then increased with increasing soil depth. The decrease in surface SOC was controlled by the decrease in agricultural fertilizer inputs, while the increase in deep SOC was caused by the inability to utilize deep soil nutrients due to the death of crop roots.

Table 2 Soil Properties Data of Different Types of Sampling Points

| Land types | Planting method | Vegetation types | 0-5cm | | 5-15cm | | 15-30cm | | 30-60cm | | 60-100cm | |
|---|---|---|---|---|---|---|---|---|---|---|---|---|
| | | | Bulk density | pH value | Bulk density | pH value | Bulk density | pH value | Bulk density | pH value | Bulk density | pH value |
| Terrace | Single vegetation | Wheat | 1.26 | 8.07 | 1.27 | 8.06 | 1.31 | 8.13 | 1.34 | 8.20 | 1.36 | 8.19 |
| | | Apple trees | 1.26 | 8.11 | 1.27 | 8.11 | 1.31 | 8.15 | 1.34 | 8.21 | 1.36 | 8.21 |
| | | Potatoes | 1.26 | 8.14 | 1.29 | 8.13 | 1.32 | 8.17 | 1.33 | 8.23 | 1.35 | 8.23 |
| | | Legumes | 1.26 | 8.10 | 1.28 | 8.09 | 1.31 | 8.15 | 1.33 | 8.22 | 1.36 | 8.21 |
| | | Maize | 1.25 | 8.08 | 1.28 | 8.08 | 1.32 | 8.14 | 1.35 | 8.21 | 1.37 | 8.20 |

| | | | | | | | | | | | | |
|---|---|---|---|---|---|---|---|---|---|---|---|---|
| | | *Robinia pseudoacacia* L. | 1.26 | 8.09 | 1.27 | 8.08 | 1.31 | 8.13 | 1.33 | 8.23 | 1.37 | 8.19 |
| | | *Pinus tabuliformis* Carr. | 1.26 | 8.11 | 1.28 | 8.11 | 1.33 | 8.15 | 1.36 | 8.20 | 1.37 | 8.20 |
| | | *Medicago sativa* L. | 1.26 | 8.12 | 1.28 | 8.11 | 1.33 | 8.16 | 1.36 | 8.21 | 1.37 | 8.22 |
| | | Vegetable | 1.26 | 8.13 | 1.28 | 8.12 | 1.31 | 8.16 | 1.33 | 8.21 | 1.36 | 8.21 |
| | Multiple vegetation | Apple tree-legumes | 1.25 | 8.08 | 1.28 | 8.08 | 1.33 | 8.14 | 1.34 | 8.22 | 1.37 | 8.19 |
| | | Apple tree-potatoes | 1.24 | 8.10 | 1.28 | 8.10 | 1.34 | 8.15 | 1.36 | 8.22 | 1.37 | 8.21 |
| Sloping land | Single vegetation | Wheat | 1.26 | 8.08 | 1.28 | 8.08 | 1.32 | 8.14 | 1.35 | 8.21 | 1.36 | 8.20 |
| | | Grassland | 1.23 | 8.06 | 1.27 | 8.06 | 1.31 | 8.12 | 1.34 | 8.17 | 1.37 | 8.18 |
| Abandoned terraces | Multiple vegetation | Apple trees and weeds | 1.24 | 8.10 | 1.27 | 8.09 | 1.31 | 8.15 | 1.34 | 8.21 | 1.37 | 8.20 |

4) The SOC content of the abandoned apple tree terraces were lower than that of the in-use apple tree terraces. Generally, this seems to be contradicted with the observed facts in the world. I think it may be collected with seasonal fertilizers addition.

Response:Your observation is very accurate. Our research findings indeed contradict general observations. We have provided an explanation for the reasons behind this result, specifically:

When the terraced fields were abandoned in this research, the SOC content of the abandoned terraces was lower than that of the terraces in use. This is caused by the short abandoned time. Abandoned terraces may have accumulated a significant amount of organic matter during their previous use. However, due to a lack of fertilization now, this organic matter is gradually being mineralized and decomposed, which reduces the soil organic carbon (SOC) content (Lal, 2004; Wiesmeier et al., 2019). In contrast, terraces that are still in use maintain higher SOC levels thanks to continual fertilization (Nardi et al., 2004). Additionally, abandoned terraces are more susceptible to soil disturbance and erosion, leading to the loss of nutrient-rich topsoil,

which further decreases SOC levels (Deng et al., 2016).Our data also shows that the surface soil organic carbon (SOC) content in abandoned terraced fields (0-15 cm) is significantly lower than that in actively used terraced fields, which may be related to higher soil bulk density, lower pH, and surface soil erosion (Table 2).

5) The key scientific issues seem to be clear. However, I didn't catch what are the main novelties of this study. Please summarized some sentences to well address them at the end of the introduction section.

Response:We would like to thank the reviewers for their valuable feedback. We have added the following content at the end of the introduction to clarify the innovative aspects of this study:

Therefore, we collected soil samples from terraced fields and slopes, including terraces with different land uses and crop types. Our focus was on three main aspects: (1) comparing the soil organic carbon (SOC) characteristics of terraces under various land use types, cropping patterns, and ecological restoration vegetation; (2) investigating the impact of terraced land abandonment on the dynamic changes of SOC; and (3) exploring how different types of vegetation influence the accumulation and distribution of SOC within terraced systems. This study provides innovative insights for comprehensively understanding the carbon cycling processes in the terraced systems of the Loess Plateau, proposing targeted management measures to promote the sustainable development of terraced agriculture and mitigate climate change.

---

## Author Comment (AC2)

RC2: 'Comment on egusphere-2024-2047', Anonymous Referee #2, 04 Oct 2024 reply

This manuscript investigates the effect of terracing on soil organic carbon (SOC) dynamics in the Loess Plateau region, with a focus on the impacts of different vegetation types (wheat, apple, grassland, trees, etc.). Given the global significance of agricultural terraces and the urgent need to develop management practices that can sequester atmospheric carbon as SOC, this topic is valuable and interesting. However, I have identified several issues that prevent me from giving a positive evaluation of this manuscript.

Response:We appreciate the valuable comments provided by the reviewers. We have taken your key points into serious consideration and made targeted revisions and improvements. We have included data on soil bulk density and soil pH, among other physicochemical properties, to further discuss the mechanisms by which terracing and vegetation types affect SOC dynamics. We added high-resolution data on soil organic carbon distribution and slope to further explore the impact of topography on soil organic carbon distribution. We have revised the abstract format according to the requirements of the journal Biogeosciences (BG). Additionally, we have thoroughly checked and corrected grammatical errors and issues related to language logic throughout the paper.

Major concerns:

The manuscript discussed the mechanisms through which terracing and vegetation types impact SOC dynamics, identified various potential factors such as water content, soil fertility, erosion reduction, roots, biomass, and crops (see sections 4.2 and 4.3). However, none of these factors were measured in this study—only SOC content was assessed for all samples (as seen in all figures and tables, which only present SOC content data). This approach significantly reduces the value of the paper and introduces considerable uncertainty in the discussions and conclusions. Could the authors include additional data in their current analysis, such as nutrients, root biomass, slope gradients, and bulk density, to better support their interpretations?

Without such data, the value of current manuscript is difficult to discern.

Response:We have carefully considered your suggestions and made additions and improvements to the relevant sections of the paper. Below are our specific responses:

You mentioned that we discussed various potential factors affecting the dynamics of soil organic carbon (SOC) but did not actually measure these factors. This is indeed a shortcoming on our part that may introduce some uncertainty. To address this, we have added data on soil bulk density and pH value, integrating this information into sections 4.2 and 4.3 to better support our arguments.

We also recognize that relying solely on SOC content data may not fully explain the mechanisms behind SOC dynamics. In sections 4.2 and 4.3, we will provide additional analysis on how soil bulk density and pH value impact SOC, strengthening our discussion.

Furthermore, we will candidly address the limitations in our data collection and measurement metrics within the research limitations section, and we commit to improving our data collection methods in future studies to enhance the reliability of our analysis.

The specific modification details are as follows:

**4.2 Effect of terraces abandonment on SOC**

As in other parts of the world, industrialization and urbanization have led to a large population flock from rural to urban areas as in China, resulting in the abandonment of a large number of productive potential farmlands (Wiesmeier et al., 2012; Cai et al., 2016). Furthermore, climate change induced extreme weather events such as drought and heavy rainfall can also accelerate soil erosion and loss of soil organic carbon in the abandoned terraces (Lal, 2004). We measured the physicochemical properties of the soil in terraced fields with different usage statuses (Table 2). The results show that the soil bulk density in abandoned terraces is significantly higher compared to the actively used ones. This increased bulk density may lead to reduced soil aeration, thereby inhibiting the decomposition of organic matter. Furthermore, the soil pH in abandoned terraces has also decreased, which may affect the stability of organic matter. However, climate change can also impact the

vegetation succession on abandoned terraces, which in turn affects the soil organic carbon dynamics (Davidson & Janssens, 2006). When the terraced fields were abandoned in this research, the SOC content of the abandoned terraces was lower than that of the terraces in use. This is caused by the limited abandoned time. Abandoned terraces may have accumulated a significant amount of organic matter during their previous use. However, due to a lack of fertilization now, this organic matter is gradually being mineralized and decomposed, which reduces the soil organic carbon (SOC) content (Lal, 2004; Wiesmeier et al., 2019). In contrast, terraces that are still in use maintain higher SOC levels thanks to continual fertilization (Nardi et al., 2004). Additionally, the abandoned terraces are more susceptible to climate change induced soil disturbance and erosion, leading to the loss of nutrient-rich topsoil, which further decreases SOC levels (Zhao et al., 2013). Our data also shows that the surface soil organic carbon (SOC) content in abandoned terraced fields (0-15 cm) is significantly lower than that in actively used terraced fields, which may be related to higher soil bulk density, lower pH, and surface soil erosion (Table 2). To produce significant environmental benefits, the land must remain abandoned for an extended period to accumulate substantial amounts of both plant biomass and the species that constitute intact ecological communities. This process can take decades to reach levels of carbon sequestration or biodiversity comparable to those of undisturbed ecosystems (Crawford et al., 2022; Poorter et al., 2016). Due to the limited water resources available in semi-arid areas, a longer natural or assisted recovery time is required. Therefore, the duration of land abandonment is a crucial factor influencing the dynamic changes SOC (Djuma et al., 2020; Badalamenti et al., 2019). In related studies in other regions, soil carbon stocks increased by 13% and 16% in cropland abandoned for 15 and 35 years, respectively (Novara et al., 2014). With the abandonment of disposal time extended, vegetation types gradually transition to grassland, scrub, and forest and the death of plants and animals return to the soil as organic matter, increasing the number of soil aggregates and further increasing the carbon content of the soil (Liu et al., 2020). Therefore, ecological restoration of newly abandoned terraces should be carried out as soon as possible. After short-term

[revised manuscript text omitted]

Your Discussion has focused on the effects of terrace abandonment on SOC content (section 4.2), but within your analysis, only 5 out of 77 sites are abandoned terraces (only considered apple tree), this is unfortunately not ideal.

Response: We are indeed aware of the limitations posed by having only five abandoned terraced field sample sites. Due to constraints in time and funding for field research, we are unable to include more abandoned terraced field sample sites in this study.

We have supplemented our research with high-resolution remote sensing imagery data to compensate for the insufficient number of field samples. However, we honestly acknowledge that this alternative method may not fully replace the data obtained from field sampling, which could affect the representativeness and reliability of the results.

We will clearly state in the limitations section of the study that the inadequate sample size may impact the assessment of the effects of abandoned terraced fields on soil organic carbon. We will strive to increase the number of abandoned terraced field samples in future research to analyze the impact of terraced field abandonment on soil organic carbon more comprehensively.

**4.4 Study limitations**

The results of the study are based on field data collected over a relatively short period of time. Due to the complexity of field conditions, the number of soil profiles in some of the comparative studies in the sampling frame design was not entirely consistent. Factors such as root biomass, fertilizer management, and tillage practices

also affect soil organic carbon in terraced areas. To compensate for the limited field sampling, we incorporated high-resolution remote sensing data into the analysis. However, this alternative method may not fully replace the data obtained from comprehensive field measurements, which could affect the representativeness and reliability of the results. Going forward, future research should strive to increase the number of abandoned terraced field samples and collect a wider range of soil physical, chemical and biological indicators. This would enable a more comprehensive assessment of the impacts of terraced field abandonment on soil organic carbon and its underlying driving factors. We need to do more work to understand the SOC characteristics of terraced agricultural areas and how to better utilize the terraces for carbon storage and realize the economic and ecological value of terraces.

You sampled 77 terraced sites but only 7 slopes as the non-terraced control. 7 slopes are under wheat and grassland, which is also not comparable as the terraced slopes. But I totally understand that in this region it's very difficult/maybe impossible to find non-terraces slopes. One solution might be trying to find reference samples from the global products with high resolution if it's possible. Or author could consider revise this paper by only focusing on the effect of difference vegetation types in terraces systems?

Response:Thank you very much for your valuable suggestions. In response to your concern regarding the insufficient number of control sites for non-terraced land, we conducted a thorough analysis and made improvements. We utilized high-resolution soil organic carbon distribution data to conduct a more comprehensive analysis of soil organic carbon (SOC) distribution within the study area. The specific revisions are as follows:

**4.1 Effect of terrace construction on SOC**

In the Loess Plateau area, the average SOC content of terraces 0-100cm is 1.4 times higher than that of sloping farmland (Table 1). Figure 5 clearly shows that the SOC content decreases with increasing depth. In Zhang et al. (2013), the SOC stock at 0-100 cm depth was 4.97 kg·m$^{-2}$ in terraces and 3.09 kg·m$^{-2}$ in sloping fields, which is

1.6 times higher than the soil organic carbon stock in sloping fields, which is consistent with the results of this study. Terracing is considered an important practice to prevent water erosion and minimize the loss of SOC (Nie et al., 2017). We can observe that areas with steeper slopes generally have lower soil organic carbon (SOC) content, while areas with gentler slopes tend to have higher SOC content (Fig. 6). This phenomenon can be explained by the fact that horizontal terraces alter the surface morphology, prolonging the surface water retention time during rainfall, which increases soil moisture in the rain-fed agricultural regions of the Loess Plateau (Xu et al., 2021). There is a positive feedback relationship between soil moisture and soil carbon (Green et al., 2019). Figure 5 shows that the soil organic carbon (SOC) content in the surface layer is significantly higher than that in the deeper layers. This may be due to the interception of precipitation by the terraced fields, which provides water for plant growth, increases plant biomass, and subsequently enhances the organic matter input into the soil. Additionally, the interception of rainfall by the terraces means that less soil fine particles are washed away, leading to an increase in the clay content of the soil. Soil clay particles have a larger specific surface area, allowing them to adsorb more soil organic carbon and enhancing the accumulation of organic carbon (Post et al., 1982). Compared to sloping land, terraces have a higher content of both clay and silt in the soil. The terraces therefore further contribute to carbon accumulation in the terraces by protecting the fine particles in the soil. In a study on the Loess Plateau, the SOC content of 0-100 cm in unterraced date palm orchards was 2.6 g·kg$^{-1}$, which was lower than the soc content of terraced orchards. This evidence further demonstrates the positive effect of terracing on soil organic carbon sequestration (Gao et al., 2017).

[Figure]

Fig.5 Distribution of slope and soil organic carbon (SOC).

(a): Slope; (b): Distribution of SOC at 0-5 cm; (c): Distribution of SOC at 5-15 cm;

(d): Distribution of SOC at 15-30 cm; (e): Distribution of SOC at 30-60 cm; (f):

Distribution of SOC at 60-100 cm.

More minor comments:

Abstract: I'm not sure if the format of abstract meets the requirement of BG journal.

Please check

Response: Thank you for your valuable feedback. I have carefully reviewed the abstract format and made adjustments based on the requirements of the Biogeosciences (BG) journal. Here is the revised abstract:

**Abstract:** Terracing is widely distributed in mountainous and hilly areas worldwide to increase grain production, control soil erosion, increase soil moisture, and improve soil quality, potentially impacting soil carbon pools. This study investigates how

agricultural activities and ecological restoration measures affect soil carbon pools in terraced areas of the Chinese Loess Plateau. We established an observation system in typical terraces and collected soil samples from 0-100 cm depth in terraces with different crops and ecological restoration vegetation. Our results show that terracing effectively increases soil organic carbon (SOC) content, with terraced cropland (7.7 g·kg⁻¹) having higher SOC than sloping cropland (4.9 g·kg⁻¹), In the 0-100 cm layer, SOC content in terraced wheat fields was 1.5 times higher than in sloping wheat fields, with the most significant increase in the top 0-30 cm. This increase is attributed to improved soil and water conservation capacity and agricultural activities. Short-term abandonment led to SOC loss, while replanting fruit trees and crops increased SOC. Our findings provides valuable insights for agricultural management and ecological restoration in terraced areas of the Loess Plateau and contributes to the development of effective carbon sequestration policies for terraced arable lands.

Line 27 changes in organic carbon content in terraces is mainly driven by the improved soil and water….

Response : This sentence does indeed have some grammatical issues and unclear structure. Here is the revised version:

Changes in the organic carbon content of terraced land are attributed to several factors. One major factor is the improvement in soil and water conservation capacity due to agricultural activities. Short-term abandonment of terraces can lead to a loss of soil organic carbon. Conversely, replanting fruit trees and crops can increase soil organic carbon content.

Line 45 is is?

Response:Thank you very much for your careful observation. There is indeed a repetition of "is" in the original sentence. We should correct this grammatical error. Here is the revised sentence:

Soil organic carbon is a key element of the global carbon cycle ( Rossel et al., 2019) and serves as a significant indicator for assessing soil quality and land productivity (Guillaume et al., 2021; Wang et al., 2012).

Line 62 maintain soil fertility and increase…

Response:We have restructured the logic of this sentence, specifically:

Agricultural terracing is a crucial landscape engineering measure to reduce soil erosion, maintain soil fertility, and increase agricultural productivity (Doetterl et al., 2012; Zhu et al., 2021), which is one of the ways to achieve sustainable agricultural development.

Line 64. conversion of terraces into slopes?? I think it's another way around…

Response:You are completely correct; there is indeed a conceptual error here. The phrase "conversion of terraces into slopes" in the original text is incorrect. It should be the other way around, namely the conversion of slopes into terraces. We will correct this mistake in the revised manuscript and reorganize the relevant content to more accurately express the role of terrace farming. Specifically:

Agricultural terracing is a crucial landscape engineering measure to reduce soil erosion, maintain soil fertility, and increase agricultural productivity (Doetterl et al.,

2012; Zhu et al., 2021), which is one of the ways to achieve sustainable agricultural development. The conversion of slopes into terraces significantly increases the cultivated area. Moreover, it helps prevent erosion problems (Arnáez et al., 2015) and effectively increases food production (Tarolli et al., 2014). Terraces are widely distributed and have created environmental benefits in countries in East Asia, the Mediterranean, and Southeast Asia (Wei et al., 2016). The terraced field construction in the Loess Plateau region of China has a long history. China has historically emphasized soil and water conservation as well as agricultural production, gradually developing and refining terracing techniques in this process. By 2005, the area had established terraced fields covering 14,790 square kilometers, accounting for 95.3% of the total arable land (Ma et al., 2015). This long history of terrace construction may influence the current processes and storage of soil organic carbon in the region. In the Chinese Loess Plateau region, a large amount of arable land has been converted to woodland and grassland through the implementation of afforestation and reforestation policies, and these measures have increased the organic carbon content of the soil (Rong et al., 2021). Many studies have shown that terracing can intercept more than 80% of rainfall runoff and sediment, and horizontal terracing can retain all rainfall to replenish soil moisture. The positive benefits of carbon capture generated by terraces come from the collection of eroded material from sloping soils. However, the conversion of natural vegetation to cropland inevitably results in a reduction of biomass and therefore a significant loss of soil organic carbon (SOC) (Aguilera et al., 2013, 2018). During the construction of terraces, it is inevitable to strip topsoil and

expose deep soil, resulting in a large amount of new subsoil covering the surface of the terraces. This severe soil disturbance may alter soil organic carbon dynamics (Sidle et al., 2006), but the potential long-term benefits of terrace construction are considerable (Chen et al., 2017). Nevertheless, many terraces are experiencing ridge damage and collapse due to a lack of maintenance or land abandonment. This not only reduces soil and water conservation benefits but potentially increases erosion and carbon emissions (Arnáez et al., 2015; Wen et al., 2020).

Line 72 sloping soils? -> soils in sloping land

Response: Thank you very much for your careful review of our article and the valuable suggestions you provided. The issues you pointed out are indeed significant, and we recognize that the original wording may have led to misunderstandings or confusion. Taking your advice into consideration, we have decided to revise this sentence to convey our meaning more accurately and clearly. The revised sentence is as follows:

The positive benefits of carbon capture generated by terraces come from the collection of eroded material from soils on sloping land.

Line 81 but also

Response:Thank you very much for carefully reviewing our article and pointing out this issue. We sincerely apologize, as this was indeed an error caused by our oversight.

The revised sentence is as follows:

This not only reduces soil and water conservation benefits but also potentially increases erosion and carbon emissions (Arnáez et al., 2015; Wen et al., 2020).

Thank you once again for your valuable feedback. We will be more cautious in our future work to avoid similar mistakes.

Line 98-99 I don't understand this sentence

Response:Thank you very much for carefully reviewing our article and pointing out this issue. We understand your confusion regarding this sentence and sincerely apologize for not expressing it clearly. We will revise and optimize this sentence to better fit the context and enhance clarity. The modified paragraph is as follows:

Nair et al. (2009) reviewed seven studies on soil carbon in tropical agroforestry systems, consistently demonstrating that agroforestry practices lead to higher carbon storage compared to conventional agricultural systems.

Line 102 By the end of 2012, there were 37,100 km2 of terraced fields on the Loess Plateau. please add a reference

Response:Thank you very much for your careful review and valuable suggestions. The issues you pointed out are very important, and we fully agree that references should be added for this data. The revised content is as follows:

By the end of 2012, there were 37,100 km2 of terraced fields on the Loess Plateau (Ma et al., 2015).

Ma, Y., Li, X., Guo, L., & Lin, H. (2015). Hydropedology: Interactions between pedologic and hydrologic processes across spatiotemporal scales. Earth-Science Reviews, 150, 201-220.

Line 110 to be clearer: we gathered soil samples from terraces and non-terraced

slopes,

Response:We fully agree with your opinion that this revision can definitely make our description of the research methodology clearer and more accurate.

Consequently, we gathered soil samples from terraces and non-terraced slopes, including terraces with varying land use and crop types, to investigate (1) the impacts of terrace construction on SOC in the Loess Plateau region and (2) the effects of different vegetation cover and tillage activities on the carbon sink capacity of terraces.

Line 147 please check this sentence

Response:Thank you very much for your feedback. We have reorganized the logic of the sentences. Specifically:

We randomly set up 84 sampling sites in the study area, comprising 77 terraced sampling sites and 7 slope sampling sites. Recognizing that crop type, cropping pattern, and agricultural abandonment all affect the soil carbon pool of terraces, we included sampling points for different cropping patterns on the terraced sites. The distribution of sampling points across different cropping patterns was as follows: 9 sampling points each for apple trees, vegetables, wheat, legumes, potatoes, and maize; 3 sampling points each for apple tree-legume intercropping and apple tree-potato intercropping. The remaining 17 sampling points were from abandoned terraces. (Appendix, Fig. S1)

Line 188 0.7 g kg-1, report sd/se along with your mean value

Response:Thank you for your feedback. Based on your suggestions, we have made revisions and additions to the original sentence. The updated content is as follows:
The SOC content of the abandoned apple tree terraces ($7.46 \pm 0.76$ g·kg$^{-1}$) was lower than that of the in-use apple tree terraces ( $8.16 \pm 1.02$ g·kg$^{-1}$), with a difference of 0.7 $\pm 1.27$ g·kg$^{-1}$.

Line 237-240 sentence is too long

Response:Thank you very much for taking the time to review our article and for your suggestion. Based on your feedback, we have optimized and revised this sentence. The revised content is as follows:

In the 0-10 cm surface layer, the SOC content under alfalfa was significantly lower than that under the two tree species. This difference became smaller in the 10-20 cm depth. However, the difference increased again in the 20-60 cm depth. At 70-100 cm, the difference between the three vegetation types became smaller, with their SOC contents converging.

Conclusion: some statements are not supported by data -> see major comments

Response:Based on the comments you provided in the "Main Issues" section, we have made revisions. We have included data on soil bulk density and soil pH, among other physicochemical properties, to further discuss the mechanisms by which terracing and vegetation types influence soil organic carbon (SOC) dynamics. Additionally, we incorporated high-resolution data on soil organic carbon distribution and slope to further examine the impact of topography on soil organic carbon distribution. The specific modifications are as follows:

**4 Discussion**

**4.1 Effect of terrace construction on SOC**

In the Loess Plateau area, the average SOC content of terraces 0-100cm is 1.4 times higher than that of sloping farmland (Table 1). Figure 5 clearly shows that the SOC content decreases with increasing depth. In Zhang et al. (2013), the SOC stock at 0-100 cm depth was 4.97 kg·m$^{-2}$ in terraces and 3.09 kg·m$^{-2}$ in sloping fields, which is 1.6 times higher than the soil organic carbon stock in sloping fields, which is consistent with the results of this study. Terracing is considered an important practice to prevent water erosion and minimize the loss of SOC (Nie et al., 2017). We can observe that areas with steeper slopes generally have lower soil organic carbon (SOC) content, while areas with gentler slopes tend to have higher SOC content (Fig. 6).

This phenomenon can be explained by the fact that horizontal terraces alter the surface morphology, prolonging the surface water retention time during rainfall, which increases soil moisture in the rain-fed agricultural regions of the Loess Plateau (Xu et al., 2021). There is a positive feedback relationship between soil moisture and soil carbon (Green et al., 2019). Figure 5 shows that the soil organic carbon (SOC) content in the surface layer is significantly higher than that in the deeper layers. This may be due to the interception of precipitation by the terraced fields, which provides water for plant growth, increases plant biomass, and subsequently enhances the organic matter input into the soil. Additionally, the interception of rainfall by the terraces means that less soil fine particles are washed away, leading to an increase in the clay content of the soil. Soil clay particles have a larger specific surface area, allowing them to adsorb more soil organic carbon and enhancing the accumulation of organic carbon (Post et al., 1982). Compared to sloping land, terraces have a higher content of both clay and silt in the soil. The terraces therefore further contribute to carbon accumulation in the terraces by protecting the fine particles in the soil. In a study on the Loess Plateau, the SOC content of 0-100 cm in unterraced date palm orchards was 2.6 g·kg$^{-1}$, which was lower than the soc content of terraced orchards. This evidence further demonstrates the positive effect of terracing on soil organic carbon sequestration (Gao et al., 2017).

[Figure]

Fig.5 Distribution of slope and soil organic carbon (SOC).

(a): Slope; (b): Distribution of SOC at 0-5 cm; (c): Distribution of SOC at 5-15 cm;

(d): Distribution of SOC at 15-30 cm; (e): Distribution of SOC at 30-60 cm; (f):

Distribution of SOC at 60-100 cm.

The SOC varies significantly in terms of the amount of plant and animal residues entering the soil and the depth of the soil under agricultural cultivation (Koga et al., 2020). The impact of agricultural activities on the surface soil levels was stronger compared to the deeper soil levels (Li et al., 2020). In this study, we observed a significant increase in SOC in terraces than in sloping lands, particularly in the 0-30 cm soil layer (Fig.6). Post-terracing, SOC sequestration in deeper soils lagged behind that in surface soils. Furthermore, the rate of SOC change was more pronounced in the surface layer (0-20 cm) compared to the deeper layer (20-100 cm). Precipitation in the region is limited and cannot replenish deep soil water, and the erosion of precipitation on the slope surface also mainly takes away the top soil layer. Therefore, the soil and water conservation effect brought by terrace construction is limited, so for the soil depth increases, this effect will become smaller. The impact of terracing on SOC sequestration diminishes as soil depth increases (Deng, Liu, and Shangguan., 2014). As soil depth increases, the water stored in the terraces cannot penetrate deeper

soils and deeper soils will maintain their properties. Therefore, the management and conservation of terrace topsoil are important to ensure local food production and enhance the carbon sink function (Li et al., 2014).

[Figure]

Fig.6 Effect of terrace construction on SOC, soil moisture, and soil grades.

(a): Variation in surface morphology by terrace construction; (b): variation in SOC content; (d): variation in soil moisture; (d), (e), and(f): variation in soil grades. The number of profiles is 9 for terraces and 4 for sloping fields. Bars denote the standard deviation of the mean.

**4.2 Effect of terraces abandonment on SOC**

As in other parts of the world, industrialization and urbanization have led to a large population flock from rural to urban areas as in China, resulting in the abandonment of a large number of productive potential farmlands (Wiesmeier et al., 2012; Cai et al., 2016). Furthermore, climate change induced extreme weather events such as drought and heavy rainfall can also accelerate soil erosion and loss of soil organic carbon in the abandoned terraces (Lal, 2004). We measured the physicochemical properties of the soil in terraced fields with different usage statuses (Table 2). The results show that the soil bulk density in abandoned terraces is significantly higher compared to the actively used ones. This increased bulk density may lead to reduced soil aeration, thereby inhibiting the decomposition of organic matter. Furthermore, the soil pH in abandoned terraces has also decreased, which may affect the stability of organic matter. However, climate change can also impact the vegetation succession on abandoned terraces, which in turn affects the soil organic carbon dynamics (Davidson & Janssens, 2006). When the terraced fields were abandoned in this research, the SOC content of the abandoned terraces was lower than that of the terraces in use. This is caused by the limited abandoned time. Abandoned terraces may have accumulated a significant amount of organic matter during their previous use. However, due to a lack of fertilization now, this organic matter is gradually being mineralized and decomposed, which reduces the soil organic carbon (SOC) content (Lal, 2004; Wiesmeier et al., 2019). In contrast, terraces that are still in use maintain higher SOC levels thanks to continual fertilization (Nardi et al., 2004). Additionally, the abandoned terraces are more susceptible to climate change induced soil disturbance and erosion, leading to the loss of nutrient-rich topsoil, which further

decreases SOC levels (Zhao et al., 2013). Our data also shows that the surface soil organic carbon (SOC) content in abandoned terraced fields (0-15 cm) is significantly lower than that in actively used terraced fields, which may be related to higher soil bulk density, lower pH, and surface soil erosion (Table 2). To produce significant environmental benefits, the land must remain abandoned for an extended period to accumulate substantial amounts of both plant biomass and the species that constitute intact ecological communities. This process can take decades to reach levels of carbon sequestration or biodiversity comparable to those of undisturbed ecosystems (Crawford et al., 2022; Poorter et al., 2016). Due to the limited water resources available in semi-arid areas, a longer natural or assisted recovery time is required. Therefore, the duration of land abandonment is a crucial factor influencing the dynamic changes SOC (Djuma et al., 2020; Badalamenti et al., 2019). In related studies in other regions, soil carbon stocks increased by 13% and 16% in cropland abandoned for 15 and 35 years, respectively (Novara et al., 2014). With the abandonment of disposal time extended, vegetation types gradually transition to grassland, scrub, and forest and the death of plants and animals return to the soil as organic matter, increasing the number of soil aggregates and further increasing the carbon content of the soil (Liu et al., 2020). Therefore, ecological restoration of newly abandoned terraces should be carried out as soon as possible. After short-term abandonment, the terraced fields showed a special change pattern at different depths in this study. SOC content first decreased and then increased with increasing soil depth. The decrease in surface SOC was controlled by the decrease in agricultural fertilizer inputs, while the increase in deep SOC was caused by the inability to utilize deep soil nutrients due to the death of crop roots.

Table 2 Soil Properties Data of Different Types of Sampling Points

| Land types | Planting method | Vegetation types | 0-5cm | | 5-15cm | | 15-30cm | | 30-60cm | | 60-100cm | |
|---|---|---|---|---|---|---|---|---|---|---|---|---|
| | | | Bulk density | pH value | Bulk density | pH value | Bulk density | pH value | Bulk density | pH value | Bulk density | pH value |
| Terrace | Single vegetation | Wheat | 1.26 | 8.07 | 1.27 | 8.06 | 1.31 | 8.13 | 1.34 | 8.20 | 1.36 | 8.19 |
| | | Apple trees | 1.26 | 8.11 | 1.27 | 8.11 | 1.31 | 8.15 | 1.34 | 8.21 | 1.36 | 8.21 |
| | | Potatoes | 1.26 | 8.14 | 1.29 | 8.13 | 1.32 | 8.17 | 1.33 | 8.23 | 1.35 | 8.23 |

| | | | | | | | | | | | | | |
|---|---|---|---|---|---|---|---|---|---|---|---|---|---|
| | | Legumes | 1.26 | 8.10 | 1.28 | 8.09 | 1.31 | 8.15 | 1.33 | 8.22 | 1.36 | 8.21 |
| | | Maize | 1.25 | 8.08 | 1.28 | 8.08 | 1.32 | 8.14 | 1.35 | 8.21 | 1.37 | 8.20 |
| | | *Robinia pseudoacacia* L. | 1.26 | 8.09 | 1.27 | 8.08 | 1.31 | 8.13 | 1.33 | 8.23 | 1.37 | 8.19 |
| | | *Pinus tabuliformis* Carr. | 1.26 | 8.11 | 1.28 | 8.11 | 1.33 | 8.15 | 1.36 | 8.20 | 1.37 | 8.20 |
| | | *Medicago sativa* L. | 1.26 | 8.12 | 1.28 | 8.11 | 1.33 | 8.16 | 1.36 | 8.21 | 1.37 | 8.22 |
| | | Vegetable | 1.26 | 8.13 | 1.28 | 8.12 | 1.31 | 8.16 | 1.33 | 8.21 | 1.36 | 8.21 |
| | Multiple vegetation | Apple tree-legumes | 1.25 | 8.08 | 1.28 | 8.08 | 1.33 | 8.14 | 1.34 | 8.22 | 1.37 | 8.19 |
| | | Apple tree-potatoes | 1.24 | 8.10 | 1.28 | 8.10 | 1.34 | 8.15 | 1.36 | 8.22 | 1.37 | 8.21 |
| Sloping land | Single vegetation | Wheat | 1.26 | 8.08 | 1.28 | 8.08 | 1.32 | 8.14 | 1.35 | 8.21 | 1.36 | 8.20 |
| | | Grassland | 1.23 | 8.06 | 1.27 | 8.06 | 1.31 | 8.12 | 1.34 | 8.17 | 1.37 | 8.18 |
| Abandoned terraces | Multiple vegetation | Apple trees and weeds | 1.24 | 8.10 | 1.27 | 8.09 | 1.31 | 8.15 | 1.34 | 8.21 | 1.37 | 8.20 |

**4.3 Effect of vegetation type and planting patterns on SOC in terraces**

Vegetation types can influence SOC by modifying the soil's physicochemical structure and altering both the input and decomposition rates of SOC (Du et al., 2022; Wiesmeier et al., 2012; Wan et al., 2019). Our data shows that there are significant differences in soil pH values under different vegetation types (Table 2). For instance, forested areas have higher pH values, while grasslands have lower pH values. This could be an important factor contributing to the differences in SOC content among various vegetation types, as pH levels influence the decomposition and stability of organic matter. Our study demonstrated that, compared to terraced fields, the SOC content of afforested land at a 0-100 cm depth was higher and that the forest litter biomass was more than that of farmland, which was the main reason for this difference. Planted forest land reduces soil temperature, soil moisture evaporation,

[revised manuscript text omitted]